# The Value of Scientific Knowledge Dissemination for Scientists—A Value Capture Perspective

**Susanne Beck** [1,2,*], **Maral Mahdad** [3], **Karin Beukel** [3] and **Marion Poetz** [1,2]

[1] Ludwig Boltzmann Gesellschaft, Open Innovation in Science Center (LBG OIS Center), Nußdorfer Str. 64, 1090 Vienna, Austria

[2] Department of Strategy and Innovation, Copenhagen Business School, Kilevej 14A, 2000 Frederiksberg, Denmark

[3] Department for Food and Resource Economics, University of Copenhagen, Rolighedsvej 25, 1958 Frederiksberg Copenhagen, Denmark

* Correspondence: susanne.beck@lbg.ac.at

**Abstract:** Scientific knowledge dissemination is necessary to collaboratively develop solutions to today's challenges among scientific, public, and commercial actors. Building on this, recent concepts (e.g., Third Mission) discuss the role and value of different dissemination mechanisms for increasing societal impact. However, the value individual scientists receive in exchange for disseminating knowledge differs across these mechanisms, which, consequently, affects their selection. So far, value capture mechanisms have mainly been described as appropriating monetary rewards in exchange for scientists' knowledge (e.g., patenting). However, most knowledge dissemination activities in science do not directly result in capturing monetary value (e.g., social engagement). By taking a value capture perspective, this article conceptualizes and explores how individual scientists capture value from disseminating their knowledge. Results from our qualitative study indicate that scientists' value capture consists of a measureable objective part (e.g., career promotion) and a still unconsidered subjective part (e.g., social recognition), which is perceived as valuable due to scientists' needs. By advancing our understanding of value capture in science, scientists' selection of dissemination mechanisms can be incentivized to increase both the value captured by themselves and society. Hence, policy makers and university managers can contribute to overcoming institutional and ecosystem barriers and foster scientists' engagement with society.

**Keywords:** value capture; scientific knowledge production; open innovation in science; subjective exchange value; open science; societal impact

## 1. Introduction

Developing solutions to handle today's challenges such as the climate crisis, demographic changes, migration, or digitalization requires the recombination of knowledge from different public, scientific, and commercial stakeholders. To this end, knowledge dissemination is a necessary condition to make knowledge accessible to relevant stakeholders. However, there are a multitude of different dissemination mechanisms, which vary in their degrees of knowledge accessibility (i.e., the number of actors that can access the knowledge) and, consequently, in their value created and captured by the knowledge-using parties (use value) and knowledge-producing parties (exchange value). Stimulating (open) dissemination beyond the boundaries of academia and, thus, increasing the use value, has become a central task for policy makers and scientific institutions (for example, as codified in universities' "Third Mission" or "Quadruple Helix" concepts) [1]. To achieve these organizational-level and ecosystem-level goals, it is crucial to recognize how individual scientists capture value from

different dissemination activities, given that this influences the likelihood of selecting a particular dissemination activity [2,3].

Although knowledge dissemination is a main and crucial part of a scientist's job, insights into strategies for systematically capturing value from scientific knowledge are, so far, mainly described in the context of university-industry collaborations and in connection with science-based entrepreneurship [4–6]. Such value capture mechanisms primarily consist of formal transfer mechanisms such as patenting, licensing, or consultancy that generate royalty fees in the short-term or long-term (i.e., monetary rewards). However, most dissemination of scientific knowledge happens through paper or book publications, conferences, or education. Since much of the current literature considers value captured in exchange for scientific knowledge to be monetary only [2,3,7,8], these mechanisms would paradoxically only indirectly lead to value captured for the knowledge creating scientists[1]. This becomes even more apparent when considering "on top" dissemination activities that currently either do not contribute to a scientist's performance assessment, or that contribute to a lesser extent in current evaluation systems (e.g., continuous education, social engagement, and knowledge transfer including open knowledge/data transfer practices). Often, such scientific dissemination activities aim to foster linkages to the general public as encouraged by universities' Third Mission efforts (e.g., public engagement, science nights, continuing education, social engagement, and dialogue) [9]. Following this current understanding, scientists engaging in such Third Mission activities are either irrational economic actors, or the application of a value capture perspective in the scientific context bears inefficiencies in explaining what value scientists receive in exchange for their knowledge dissemination activities. Subsequently, the question is why scientists apply dissemination mechanisms that go beyond their performance evaluation scheme (e.g., publications versus social engagement). Therefore, in this paper, we want to explore the following research question: How do individual scientists capture value from their scientific knowledge production and dissemination activities?

Drawing on the knowledge-based view, scientific knowledge is considered as value created [10] on an individual-level by one or more scientists [3]. However, the value scientists receive in exchange for their scientific knowledge [2,3], i.e., its dissemination, remains unclear beyond monetary rewards. Our point of departure to understand how scientists capture value from scientific knowledge dissemination is, therefore, to investigate what scientists perceive as their "realized exchange value" of scientific knowledge dissemination.

Moreover, knowledge production requires the investment of resources and is associated with uncertainty—i.e., whether value can be captured at all [2,3,7,11]. One aspect driving scientists' willingness to engage in the value creation process is the anticipated exchange value [2,3]. However, knowledge regarding what is considered valuable by scientists remains scarce [5,12,13]. Therefore, and, in order to better understand scientists' selection of a particular dissemination mechanism, it is essential to recognize why the anticipated exchange value is considered (sufficiently) valuable to produce the knowledge in the first place.

To explore value capture mechanisms from scientific knowledge production and dissemination, we applied an inductive-deductive qualitative research design, gathering data from two comprehensive workshops with scientists in Denmark and Austria, each lasting several days (totaling more than 60 h of observing 30 scientists' directly and indirectly describing their anticipated and realized exchange values). In addition, we conducted eleven semi-structured interviews with scientists with diverse disciplinary and nationality backgrounds (totaling approximately 15 h of interviews). Based on our findings, we propose that the value scientists receive from disseminating their knowledge consists of an objective (mostly monetary) and an additional subjective (non-monetary) dimension, which

---

[1] We consider salary to only be indirectly (not directly) related to the actual value creation process (i.e., knowledge dissemination activities). We acknowledge throughout the manuscript that publications, conferences, etc. contribute to a positive performance assessment of the scientist, consequently improving job opportunities in academia, and, thus, receiving salary. This however, has a diminishing marginal utility, once the scientist has a tenured position.

are perceived as valuable due to different reasons. Hence, our results add to the understanding of how and why scientists capture value from scientific knowledge dissemination. First, the subjective value comprises of outcomes such as social recognition, reputation, or social acceptance. Second, these outcomes are considered valuable due to scientists' individual needs, such as the struggle for academic survival (e.g., position), ego-identity needs (e.g., social desirability), as well as the desire to make a societal impact. In contrast, objective value mainly comprises of measurable outcomes (e.g., monetary rewards). In essence, the results show how scientists' needs are met by the objective and the subjective exchange value explaining scientists' willingness to disseminate and their selection of dissemination mechanisms. Moreover, they contribute to understanding what drives scientists' engagement in the scientific knowledge production in the first place.

These individual-level findings hold meaningful implications for both the organizational-level and policy-level. First, we contribute to improving our understanding of scientists' value capture processes from different scientific knowledge dissemination activities. We identify what scientists consider valuable beyond the monetary reward and, hence, add to an important and still under-researched aspect, i.e., the value of scientific knowledge dissemination from the scientists' perspective [14,15]. Second, our findings contribute to the discussion surrounding how policy makers, research funders, university managers, or institutions can incentivize scientists' engagement in Third Mission or Quadruple Helix activities, which aim to achieve high societal impact by fostering knowledge transfer between academia and society [1,16]. The opening up of knowledge dissemination transforms scientific knowledge into a commodity good [6,17,18], with substantial consequences for the value captured by society (i.e., the use value), due to an increased number of knowledge users facilitating new knowledge generation [2,3,6,15,19,20]. In other words, by disseminating scientific knowledge, other actors are able to use and recombine it and, thus, create additional knowledge (i.e., value) to address relevant societal challenges. Therefore, such (open) dissemination activities, that go beyond the current discipline-dependent evaluation schemes in academia are able to increase both, the value captured by the public and the value captured by the scientists. Third, we add to the small body of literature that studies individuals as the unit of analysis in the science context and how their cognitive and emotional behaviors play a role in the context of open scientific knowledge production and dissemination [3,21,22]. Not only do we underline the importance of paying attention to macro-level factors, but we also highlight the importance of considering the micro-foundations of scientific knowledge production, since individuals (e.g., scientists) are important decision makers. Understanding what scientists consider valuable allows policy makers and university managers to optimize incentive schemes to stimulate scientists' individual selections of dissemination activities that achieve higher societal impact.

## 2. Theoretical Background

This section starts by briefly reviewing the literature on value creation and value capture in the context of scientific knowledge production. In this vein, use value and exchange value are outlined. Following this, we describe the value creation and value capture processes in this context and identify inefficiencies that lead to our exploratory research question.

### 2.1. Creating and Capturing Value from Scientific Knowledge Production

Understanding value creation and value capture has received considerable attention in management research [2,3,7,23]. The underlying assumption is that innovations (i.e., new products, services, or processes) create value that is distributed among different stakeholders [24,25]. Thereby, value creation and value capture must be understood as interdependent processes [2,7]. To appropriate value from its innovations, firms need to apply value capture mechanisms (e.g., licensing, patenting, and sales) that allow them to realize innovation rents from one particular and subsequent innovations [20,26].

Value cannot only be created on an organizational level, but also on an individual, collective, and societal level [3,10]. Building on the understanding of the knowledge-based view, knowledge from individuals is considered an important resource in creating value [10]. Accordingly, scientific

knowledge production has been considered as a value creation process in previous work [5,10,18]. In the following, we consider scientific knowledge as the value created by an individual scientist [3,5]. Thus, we focus on value creation on an individual level, in the context of scientific knowledge production.

Capturing value from scientific knowledge production has received comparably less attention. One way to capture value from scientific research for society, the economy, and the scientists themselves [2,3,18] is its transformation into innovation. This transformation has been addressed by two major literature streams. First, university-industry collaboration, and, second, science-based entrepreneurship [4]. Whereas, in the first case, scientific knowledge (value created) is exchanged with another economic actor (e.g., a firm). In the second case, it might be transformed into an innovation by the scientist her/himself. Figure 1 graphically depicts the theoretical understanding of value creation and value capture, in general, and in the context of university-industry collaboration and science-based entrepreneurship in particular.

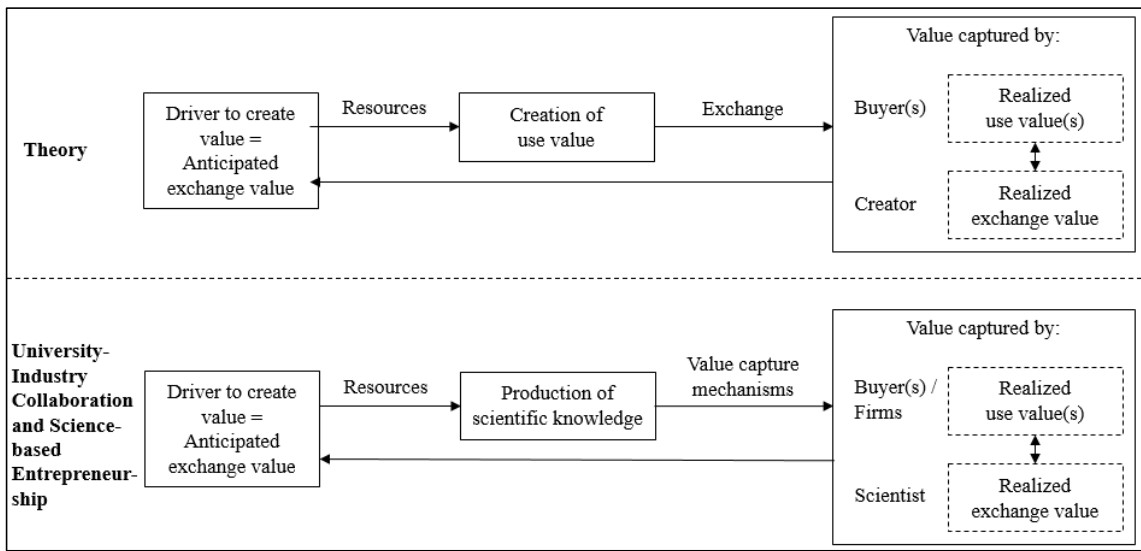

**Figure 1.** Graphical summary of the process of value creation and value capture in general and from scientific knowledge. Source: own illustration adapted from Bowman and Ambrosini (2000) [2].

Value is captured by two types of actors when it is exchanged. While, in the case of an innovation, the exchange occurs between buyers of a new product or service and its producers [2], the buyers might be firms that decide to collaborate with a scientist in the context of scientific knowledge. However, to describe value capture mechanisms and strategies first requires the definition of value. Building upon Teece's model [24,25] of the overall value captured by an innovation, Bowman and Ambrosini [2] differentiate between use value and exchange value. While the latter is the (monetary) price paid to obtain a good, the first is the buyer's surplus. The surplus describes the comparison buyers make between products, their needs, and the feasibility of other offerings such as comparisons that resource suppliers make between the deal with the firm and possible other deals. While these authors focus on the organizational level, Lepak, Smith, and Taylor [3] broaden this understanding by accounting for a societal and individual-level perspective. Value creation, thereby, "depends on the relative amount of value that is subjectively realized by a target user (or buyer) who is the focus of value creation—whether individual, organization, or society—and that this subjective value realization must at least translate into the user's willingness to exchange a monetary amount for the value received" [3]. Thereby, the value created must be a contribution that is perceived to be valuable by members of a target group [3,10]. Hence, the value created must exceed the perceived utility of any other alternative presented to the target group by either lowering the cost or creating a higher value. However, while the use value is considered to be subjectively perceived, the exchange value lacks such a subjective aspect [2].

The exchange of scientific knowledge happens through different knowledge dissemination activities. Typically, scientific knowledge dissemination happens through scientific publications, conference presentations, book presentations, interviews, and so forth. Thereby, the selection of a dissemination activity is influenced by field-specific norms. Commercial value capture mechanisms are also applied by scientists, across all fields, that allow them to commercialize their knowledge such as patenting, licensing, consulting, or academic entrepreneurship [4,27]. In the context of university-industry collaboration and science-based entrepreneurship, this means that the right to use the knowledge is given (e.g., through licensing, consulting, and patenting) in exchange for money. Thereby, whatever utility the buyer perceives is the uniquely realized use value. This use value can differ for any actor who uses the scientific knowledge. The monetary value received by the scientists can be considered as a further realized exchange value. Hence, in the case of a publication, every reader realizes an individual use value, as well as the publisher who normally owns the rights. In this case, the realized exchange value is the royalty fee the scientists get from the publisher, based on the sales of the publication, if any[2]. In the case of a licensing-deal, the firm that licenses the scientific knowledge can create new innovations (use value in terms of future realized exchange value), while the scientists receive the licensing fees paid by the firm (exchange value). However, whether a scientist is willing to engage in the value creation process (i.e., knowledge production and dissemination) in the first place, depends on the anticipated exchange value [3] (i.e., the anticipated value to be captured by the scientist and not only on the pure ability to engage).

Value capture mechanisms, therefore, describe actions that allow scientists to capture exchange value from their scientific knowledge production and dissemination. These mechanisms can be structured, according to their level of formalization [23]. Thereby, formal mechanisms include but are not limited to patenting, collaborative research, consultancy, or licensing and informal mechanisms describe networking activities or ad-hoc advices for practitioners [4]. By applying these mechanisms, scientists are able to realize the exchange value from the disseminated scientific knowledge.

However, only a fraction of the dissemination mechanisms for scientific knowledge allow the scientist to capture such an exchange value. Scientific knowledge is a durable public good [18]. Its dissemination is a necessary condition for the exchange and recombination of information [18] and, hence, for the realization of use value and exchange value [2]. Considering that some dissemination activities are associated with a lower anticipated exchange value for the scientist, the question arises regarding why such knowledge dissemination activities are used at all. This leads to the following dilemma. While disseminating scientific knowledge to more users would increase the use value (and, thus, the overall value captured), it does not necessarily increase the scientist's exchange value, which represents a low incentive to apply these mechanisms.

### 2.2. Theoretical Framework for Analyzing the Dilemma

Despite commercialization through university-industry collaborations and science-based entrepreneurship, scientists most commonly disseminate their scientific knowledge through publications, conferences, or teaching. Understanding the exchange value as monetary rents in exchange for the scientific knowledge leads to a paradoxical situation. Such dissemination strategies lead to no or only a very limited exchange value. However, the anticipated exchange value needs to exceed a critical threshold for scientists to be willing to (further) create scientific knowledge in the first place and, thus, create use value for other individuals, organizations, and society. Consequently, scientists either act irrationally because they engage in a value creation process where the costs (i.e., the required effort) exceed the anticipated exchange value, which leads to self-destruction. Or the exchange value for scientific knowledge consists of more than monetary rewards.

---

[2] In relation to the first footnote, a tenure position and salary are considered additional values (indirectly) captured through classical dissemination activities.

The value of scientific knowledge has received considerable attention [5,6,17,18]. Most authors have focused on the description of the (realized) use value of scientific knowledge. Hence, they argue what value applied vs. basic scientific knowledge has for society and organizations [17], or describe why economic actors invest in the creation of scientific knowledge [18]. One pioneering exception is Dedrick and Kraemer [5] who describe how value creation by science-based innovation is distributed among all stakeholders—including the national ecosystems and the scientists. They point out that awarded prizes and prestige can be considered rewarding for scientists.

However, knowledge about what is considered valuable by scientists remains scarce [5]. The largest body of research focuses on the challenging environment for young scientists (see, for example, a special issue of the Journal Science in September 1999). It is no secret that the career decision to stay in academia is often related to several sacrifices such as job insecurity due to short-term employment, limited work-life balance, lower average wages, above-average working hours, and consequently also a rather hostile environment for family development, especially for female scientists [28–30]. Therefore, what drives people to engage in scientific knowledge production? A few studies on scientists' work motivation [12,31–33] provide initial answers. For example, Gibbs and Griffin [33] found that the main reason for staying in academia is the flexibility and freedom to research. Furthermore, the ability to engage in externally focused values (e.g., improving the societal status quo) were mentioned, as well as the influence on students. Combining these insights with the value capture perspective, we propose that the exchange value also consists of a non-monetary component.

In the following, we, therefore, want to explore what scientists consider as a desirable exchange value, i.e., what they want to receive in exchange for disseminating the scientific knowledge that they have produced, and what value capture mechanisms (i.e., dissemination activities) they apply for doing so. Identifying what is considered as a desirable exchange value also allows us to understand why the anticipated exchange value sufficiently drives the scientist's willingness to further engage in value creation processes (i.e., scientific knowledge production). Accordingly, our exploratory research aims to address the following research question: How do individual scientists capture value from their scientific knowledge production and dissemination activities?

Figure 2 depicts the process of value creation and value capture with the red circles highlighting the foci of the study. We first want to explore, what mechanisms (i.e., dissemination activities) scientists apply to capture value from their knowledge production. Second, what value is captured by the scientists, and third, why do scientists consider the realized exchange value valuable.

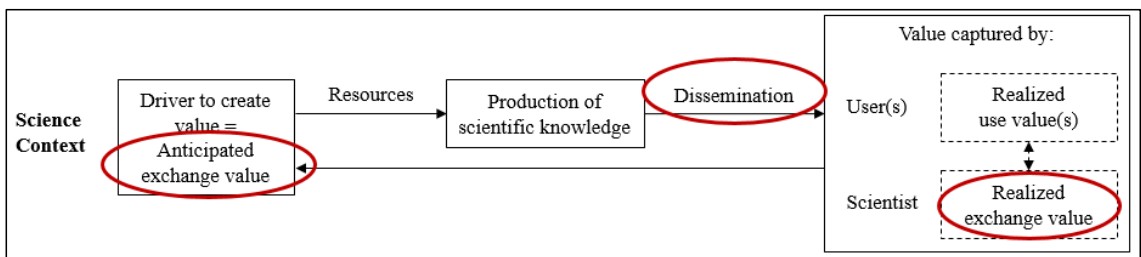

**Figure 2.** Theoretical conceptualization including the foci of this exploratory study. Source: own illustration adapted from Bowman and Ambrosini [2].

## 3. Methodology

Due to the exploratory nature of this study and the previously mentioned research question, a qualitative approach was applied to gather in-depth data and rich information on the phenomenon [34]. Qualitative approaches are known to be particularly useful for understanding the theory underlying the observed relationships in data [35]. In our case, exploratory research is considered to be the best option, since the inner content of value creation and value capture processes—what is happening in a real scientist's life—is an underexplored research domain that potentially shapes a new understanding of the phenomenon [36]. Therefore, given the "how" nature of the research question and the focus on

underlying factors associated with value creation and value capture in science rather than studying them in isolation, a qualitative study with multiple phases of data collection is required [34]. Moreover, an inductive-deductive approach was chosen for this study. First, we want to inductively explore how scientists capture value from their knowledge production and dissemination activities. Second, we use value capture theory to make sense of the data and embed our findings. Using this mix enabled us to (1) make the best use of our empirical data (i.e., let the data speak for itself, (2) incorporate pre-existing theories that study this phenomenon, and (3) enrich pre-existing theory by adding novel explorations and interconnecting elements of other theories.

### 3.1. Data Collection and Research Context

Data was collected during 2017, using two inquiry techniques over three phases (see Figure 3).

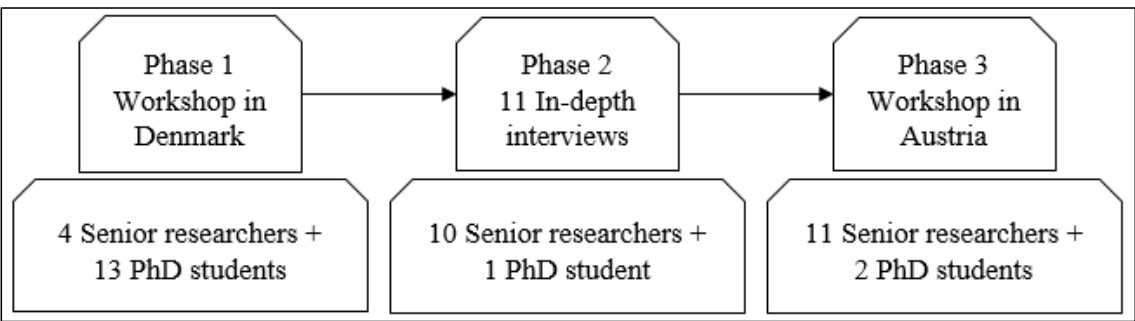

**Figure 3.** Three phases of data collection. Source: own illustration.

*Workshops:* Two workshops aimed at providing participants with frameworks and tools to develop and implement mechanisms and processes of how to capture value from their scientific knowledge. The workshops were meant to be an opportunity for participants to work on their own institutes' future approach to value capturing by developing knowledge, skills, and competencies.

*In-depth interviews (11 scientists):* The interviews aimed to better understand the process of value capture and to take a deep dive into different mechanisms, antecedents, and outcomes, from a scientist's perspective.

*Phase 1:* The first workshop was designed to inform and educate participants about value capturing in science and it helped the research team to get a better understanding of the phenomenon of value capture in science from the scientists' perspective. The information gathered in the first workshop helped to construct an interview guideline for the second phase, based on a deeper and a more practical level of understanding the phenomenon.

*Phase 2:* Building upon phase 1, we invited future participants of the second workshop to participate in an in-depth interview prior to their participation. Semi-structured interviews were conducted, using an open-ended interview protocol. The semi-structured interviews allowed informants to offer their comments freely, which allowed us to collect in-depth and field-specific insights. In drafting the interview questions, we focused on mechanisms extracted from the literature, as well as value creation and capture theory. However, as is common in explorative research, new factors and mechanisms started to reveal themselves during the interviews.

In total, we conducted 11 interviews with scientists, each took between 1 h to 1.5 h (see Table 1 for interviewees' information). Interview participants were scientists from different fields, working in different institutes of a large research organization with a thematic focus in medicine, the life sciences, social sciences, and cultural studies. Participants originated from seven different countries including Austria, Belgium, Bosnia, Herzegovina, Hungary, Italy, Poland, and the UK. The research organization is one of the largest research institutions in Austria, with 30% of its budget being publicly funded.

**Table 1.** Interviewee characteristics.

| Interviewee | Position | Research Field | Years after PhD |
|---|---|---|---|
| **A** | Administrative head | Health/life | 5 |
| **B** | Group leader | Health/life | 16 |
| **C** | Senior scientist | Social sciences | 8 |
| **D** | Scientist/PhD | Health/life | Last year PhD |
| **E** | Postdoc | Health/life | 3.5 |
| **F** | Group leader | Health/life | 8 |
| **G** | Group leader | Health/life | 15 |
| **H** | Group leader | Health/life | 13 |
| **I** | Key researcher | Humanities | 3 |
| **J** | Key researcher | Humanities | 7 |
| **K** | Group leader | Social sciences | 3 |

Prior to interviews, interviewees were informed that the questions would mostly focus on their individual dissemination activities. However, while the questions related to rewards brought up in accordance with each dissemination mechanism, the categorization of the type of reward was done in the analysis phase.

The interview guideline consists of three sections. In the first part, the interviewees discussed in detail the value capture mechanisms they currently employ, before going into more detail with the antecedents of implementing such mechanisms and the expected outcomes. These open questions addressed but were not limited to (a) participants unique careers (projects, research line, perspectives, experiences, and technical details of their research), (b) identifying mechanisms and elaborating on each mentioned mechanism in detail (when, experiences, successes, challenges before, during, and after implementing each mechanism), (c) questions considering different mechanisms mentioned earlier, such as "Why do you consider the [mechanism] to be valuable?", "What do you perceive as satisfactory about the [mechanism]?", "How do you think scientific and non-scientific communities evaluate the [mechanism]?" and "In your opinion, can you measure the value of the [mechanism] and if yes, how?".

*Phase 3:* In the last phase, the research team discussed the results of the interviews with the participants and validated the main understanding of value capture processes in science. The topics discussed with the participants in this phase were: (a) capturing value from science, (b) Open Innovation search and collaboration approaches to creating and capturing value from science, (c) intellectual property (IP), IP rights, and strategies in Open Innovation and Open Science, (d) opportunities, risks, and contingency factors related to applying Open Innovation/Open Science as a scientist, (e) identification and selection of external partners for commercializing science, (f) opportunities and challenges involved in partnering with externals, (g) the role and value of tech-transfer offices in supporting the commercialization of science, (h) working with/using intermediaries and platforms for the commercialization of science (e.g., scientists as suppliers to platform challenges), and (i) good-practice examples and case studies related to external partnering in the commercialization of science.

The data from both workshops was collected and analyzed in a systematic way by means of observation (e.g., participant's presentations and flip charts presenting value capture strategies for their research). The workshop data was triangulated with the interview data. In total, we collected 60 h of the workshop material and 15 h of in-depth interviews with scientists (approximately 300 pages of transcribed raw data).

### 3.2. Data Analysis Procedure

The transcribed data was then processed to provide a clean case for each participant. The initial analysis was mainly conducted by two researchers, based on triangulation of the data sources (workshops, documents, observations, and interviews) for each scientist. The first round of analysis was structured along the lines of the guideline used for data collection. In other words, we categorized



the individual answers, according to the thematic open-ended questions. This open and inductive coding approach contributed to obtaining a comprehensive and general picture of value capture mechanisms in science. All transcripts were reread multiple times with the following questions in mind: what mechanisms are used for capturing value from research? Why and how were these mechanisms used? This step was done with the least consideration given to predefined concepts and categories.

The second round of analysis contained a more detailed and analytical approach for each transcript. By iteratively analyzing data, literature, and concepts, different categories began to emerge [37]. This step helped us conceptually refine and connect each identified category to relevant contexts.

We used descriptive codes to identify and cluster data related to each existing and emerged concept. We then drew on a set of theoretical concepts that reflect the interplay between the main emerging concepts such as value, human behavior, motivation, and action.

Since the main purpose of our research was opening the black box of capturing value from scientific knowledge production and dissemination, we started to interpret what we considered to be valuable for scientists. Lastly, in the last analytical task, existing and emerging code concepts were categorized and shaped patterns related to different stages of value capture processes in science.

### 3.3. Validity and Reliability of the Data

Although our interviews were selected from different fields, with different nationality backgrounds, the issue of generalizability has always been present in doing our qualitative research. For example, while the scientists were from different fields, the sample comprises a smaller fraction of scientists from social sciences and humanities (each $n = 2$). It is, therefore, worthwhile to mention that the main purpose of our study is to broaden theory and reflect on the phenomenon rather than generalizing from the sample to the population. In addition, pilot interviews were conducted to construct a secure and reliable basis in terms of content and duration for the formal data collection process. Lastly, one of the two data analysts did not participate in the data collection process and started the analysis from raw transcripts. This provided a non-biased interpretation of the raw data, which was aligned with the first analyst's interpretations.

## 4. Analysis and Results

This section first highlights the design of value capture mechanisms in relation to their monetary and non-monetary outcomes. We then move beyond these mechanisms by shedding light on how they contribute to the value that scientists capture and "why" these mechanisms have been used by scientists as antecedents of value capture mechanisms. Our data indicates that dissemination mechanisms in science can be considered as value capture mechanisms. This is due to the characteristics of the exchange value that is realized by disseminating the scientific knowledge. Our data indicates that the realized exchange value is not only of monetary but also of a non-monetary nature. The non-monetary exchange value is considered by scientists as valuable due to the scientists' needs pyramid. Therefore, we open three black boxes of the scientists' value capture process: their dissemination mechanisms, their realized exchange value, and the underlying reasons why the value captured is considered valuable. However, since the nature of our analysis is explorative, some facets might appear more often than others during the open coding process. This does not necessarily imply a higher importance.

### 4.1. Mechanisms for Value Capture from Scientific Knowledge

Analyzing our dataset, we found formal and informal sets of value capture mechanisms (i.e., dissemination mechanisms) employed by scientists to capture value from their research. Formal sets of mechanisms can be identified as mechanisms that have a naturally formal structure. By formal structures, we mean the employment of dissemination mechanisms that bring monetary and non-monetary outcomes to individuals, institutions, and society. Our interviewees reported sets of formal mechanisms such as patents, publications, conferences, and teaching. These formal mechanisms

were found to have both monetary and non-monetary outcomes for scientists. However, it is worth noting that formal mechanisms are discipline-dependent.

According to our interviews, it is institutional-level factors that influence the decision to choose a formal (vs. informal) mechanism in life and health science institutes. By contrast, in the humanities and social sciences, this decision seems to be more flexible and more influenced by individual-level factors.

> *"Towards the research community, they are disseminated almost exclusively in conference talks, article publications, book publications, and book reviews, so I guess literary studies produces literature. Then, to the non-research community, we do things like book presentations, or exhibitions in museums."* (Interviewee I)

> *"[we disseminate] by publications, obviously. Then going to conferences. [ . . . ] We have the aim that everybody is able to go to the major conferences, and to present. [ . . . ] We always published consensus manuscript, as a result of this conference, like our recommendation how to classify a disease, what a stem cell is, or sometimes to recommend certain modifications of the standard treatment."* (Interviewee A)

> *"[...] If there is a new method, which can be patented, and it can be used later by the scientific community, I think it can be interpreted as if it was directed to the scientific community."* (Interviewee B)

Publication is the most frequently used formal mechanism to capture value from research for two reasons. First, because of the scientists' position in the scientific network and, second, because of the indirect monetary exchange value this mechanism creates for scientists. Indirect means that this mechanism brings about a better position for scientists in their scientific community, which promotes and enhances career opportunities.

In addition to formal sets of mechanisms, scientists reported that they disseminate their knowledge by employing various informal mechanisms. Informal mechanisms are identified as being driven by an informal structure or no-structure. Informal structures are structures that are encouraged by the scientists' institutions, but they are not predetermined tasks of scientists. Mechanisms with no structure as their basis are solely driven by the scientists themselves with no involvement from their institutions or their environment.

> *"Yesterday I got an invitation for this Science Slam in November. I should also present my project there; let's see if I can realize it."* (Interviewee D)

While formal sets of mechanism result in monetary and non-monetary outcomes, our data shows that informal mechanisms to capture value from scientific knowledge production have mostly non-monetary outcomes. Non-monetary outcomes are outcomes that have a mostly indirect impact on scientists' survival in academia.

Table 2 shows various formal and informal mechanisms that are used to capture value from scientific knowledge and their monetary and non-monetary outcomes for the producer of the scientific knowledge. For example, while the primary (**) outcome of patents is monetary (e.g., license is purchased), it also includes a secondary (*) indirect non-monetary outcome (e.g., future career opportunities). To give another example, while the primary (**) outcome of media use is non-monetary (e.g., visibility by public organization), it does not have a monetary outcome (-).

**Table 2.** Value capture mechanisms in science.

|  |  | **Monetary** | **Non-Monetary** |
|---|---|---|---|
| **Formal** | Patent | Direct ** (e.g., license) | Indirect * (e.g., career opportunities) |
|  | Publication | Indirect * (e.g., career promotion) | Direct ** (e.g., career promotion) |
|  | Conferences | Indirect * (e.g., career promotion) | Direct ** (e.g., network recognition) |
|  | Teaching | Direct ** (e.g., pay check increase) | Indirect * (e.g., career promotion) |
| **Informal** | Collaboration | - | Direct ** (e.g., Network recognition) |
|  | Book presentation | Indirect * (e.g., career promotion) | Direct ** (e.g., Social recognition) |
|  | Media | - | Direct ** (e.g., public organization visibility) |
|  | Public lectures | - | Direct ** (e.g., Social recognition) |
|  | Patient visits | - | Direct ** (e.g., Research ideas) |
|  | Science nights | - | Direct ** (e.g., Social recognition) |

Note: Asterisks indicate the strength of the relationship. ** Primary influence, * secondary influence. The types of outcome presented in the table are not exhaustive.

## 4.2. Exchange Value from the Scientist's Perspective

Our results indicate that the exchange value in science seems to be more complex than previously assumed in the literature on the commercialization of scientific knowledge. Although it mirrors the typical structure of value capture mechanisms in terms of monetary outcomes, new insights into what scientists perceive as sufficient to engage in knowledge production (i.e., value creation) and dissemination evolve. Besides the monetary outcomes that we consider as the objective exchange value, scientists recognize non-monetary-rewards that we perceive as the subjective exchange value (see Figure 4). Narratives of scientists' value capture mechanisms in their specific context are, therefore, considered to represent a dynamic structure that weaves a subjective exchange value together with an objective exchange value. These two elements combined indicate what individuals consider as a desirable exchange value when implementing any mechanisms.

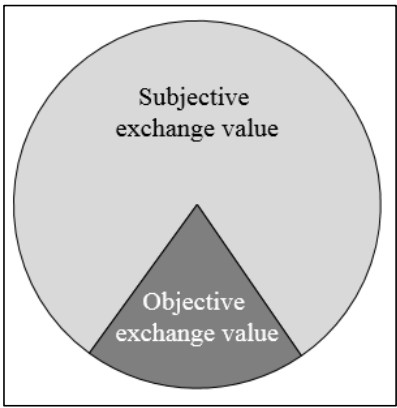

**Figure 4.** The objective and subjective part of the realized exchange value in science. Source: own illustration. Please note: the different sizes of the two fields illustrate the result from our study that scientists tend to receive a higher subjective than objective exchange value from disseminating their research. The fraction of the subjective versus objective value, however, does not indicate a specific ratio.

Our study indicates that subjective exchange values are a prevalent but unconscious part of the bigger picture of exchange value structures in science. This means that when scientists perceive value resulting from an action, they simultaneously acknowledge the subjective nature of the value. This expands the current understanding of value and its monetary nature in the science context. Hence, scientists' desire to capture value from their knowledge dissemination using formal and informal

mechanisms could be considered as a recognition of the importance of satisfying their needs. These needs are ultimately fed by the subjective value rather than the objective value they receive.

### 4.2.1. Scientists' Objective Exchange Value

Based on our data, we identify a comparably small proportion of the exchange value that determines those outcomes that can be objectified into monetary rewards. Objective values are those related to the measurable output they receive from implementing value capture mechanisms, such as increase in salary, career promotions, and research funding.

> *"[ . . . ] then, of course, it promotes the career, because you need publications in order to be able to apply for additional funding, or, in this case, for the prolongation of the cluster, for example, and then depending on the topic. Again, it is the contribution to the scientific field, so that you gain recognition, but you are also able to promote the work of others that can build up on your work." (Interviewee A)*

> *"Whatever draws attention to your research helps you because these are the things that are quantified, citations [...]" (Interviewee C)*

It's worth noting that, in line with the theoretical foundation outlined in value capture studies, our participants never mentioned basic salary as a monetary reward of their dissemination activities, but they did perceive salary increases or research funding as rewards.

> *"[...] if you get a grant, then you get an extra piece of that to support your salary. So, it translates directly. It is worth getting a grant because you are going to earn more [...] salary schemes that are in Austria are absolutely non-motivating, so I basically am motivated and propelled by my love of science and intellectual curiosity." (Interviewee I)*

We found that individuals received less objective value than they had expected when capturing value from employing formal and informal mechanisms. However, there still might be a subjective judgement on what is perceived as an objective value for scientists.

> *"For example, if you are looking for a new job, or if you are writing proposals, the reviewers would see if you are really good. If you fit into this project, if you are the right person to work on this project. And then they look in the publications." (Interviewee E)*

### 4.2.2. Scientists' Subjective Exchange Value

In the context of science, the subjective exchange value that individuals receive in exchange for disseminating their research is driven by cognitive and socio-psychological factors. The concept of subjective exchange value, however, is a broader term for non-monetary rewards resulting from a dissemination action in academia. Subjective value is, by nature, something positive (e.g., 'feeling of satisfaction', confidence, or pride) [38]. In our study, we found that it is mostly the subjective exchange value, or the subjective judgement of an objective value that satisfies scientists' cognitive and socio-psychological needs. Subjective value can be seen as the best available intuition about an objective action [38]. Therefore, it is not surprising to see that objective actions (formal and informal mechanisms) are first evaluated through a subjective exchange value. For example, one's willingness to appear on social media feeds the scientist's ego-identity status, which, in return, results in the feeling of satisfaction and of being recognized.

> *"With the general, how to say, environment, the feeling of the public towards your research, if this research is important for the well-being of the people, or if this research is just important for itself. Of course, it's always much better, if the people feel, and know that in the end, there will be something that affects their lives, or our lives in this case." (Interviewee A)*

*"I think that is very satisfactory ... Then to get responses, and yeah, visibility I think is very satisfactory. It's a requirement. So, you are judged based on your publications. Whatever you have published is kind of yours, so to say. So, your publication list will always be your publication list. It is kind of like your output, your personal out of your personal value. It is the way to sell yourself, of course, to people. This is how people are going to evaluate you, based on what you published." (Interviewee G)*

Based on our data, we, therefore, argue that implementing dissemination mechanisms leads to capturing objective as well as subjective value. Moreover, scientists' perceived subjective value outweighs the objective value when implementing any dissemination mechanism. In the following, we explore why the previously mentioned subjective and objective exchange value is considered to be a sufficient driver for scientists to engage in value creation processes.

### 4.3. Scientists' Need Pyramid

We go beyond describing research dissemination mechanisms and value capture in isolation and address why the realized exchange value drives individual scientists to engage in knowledge production and dissemination. We identified three categories of antecedents that drive mechanism selection, namely: survival in academia, ego-identity status validation, and societal impact. These needs form a pyramid in our results, labeled as the "scientists' needs pyramid" (see Figure 5). In the pyramidal form, the different sizes are intended to express different amounts of the exchange value required to satisfy these needs.

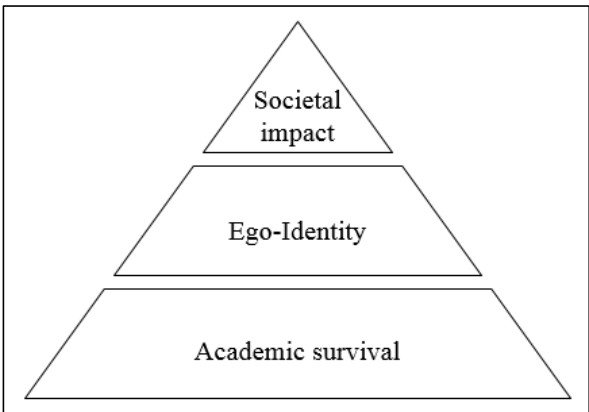

**Figure 5.** Scientists' needs pyramid. Source: own illustration.

#### 4.3.1. Academic Survival

Our data show that the main driver for producing and disseminating knowledge is survival. We found scientists' first and most important driver to use both formal and informal strategies is to survive in their academic life and continue research in their core area of interest. The desire to meet academic career goals serves as the basis of value capture from scientific knowledge (e.g., through research funds or promotion). For example, disseminating knowledge by means of attending conferences secures the individuals' unique position in their network and, therefore, signals to competitors and enhances individual bargaining power when it comes to opportunities. In the context of science, considering individuals as a separate entity explains how this strategy creates an isolation mechanism in the individuals' own network.

*"Well, the conferences are something really, really big and we do not have articles despite maybe the first idea would be to connect it with exhibitions. [ ... ] It is really an important moment for the life of a scholar. In a conference, my research is evaluated by my colleagues, and also, this is the situation, this is the moment where I can make relationships with other colleagues, other scholars.*

*Maybe organize another conference in two years, or a collaboration, or a book together. Conferences are absolutely one of the most important aspects of our research, absolutely." (Interviewee J)*

*"Sometimes it is very valuable to have international partners. Also, in some cases, like we have also been able to found societies, for example, that are international societies, for example placenta stem cells. We kind of bring the community closer, so to say. There are few people working in this field worldwide, it is a way to bring them closer together. Then this is a nice way of meeting them again at conferences or making our own meetings and conferences." (Interviewee G)*

Most of our interviewees use publications as their main knowledge dissemination mechanism. However, they do not primarily aim to share their research insights but rather use publications to make themselves identifiable to their peers and, thus, generate career opportunities.

*"It is always good to be recognized by the simple fact that somebody considered your work valuable of publishing, and we, of course, look into it to publish either in top journals [ . . . ] then, of course, it promotes the career, because you need publications in order to be able to apply for additional funding, or, in this case, for the prolongation of the cluster, for example." (Interviewee G)*

Interestingly, the story is similar for informal mechanisms. One interviewee reported the ultimate driver for presenting a book to the public is not only about getting noticed by the larger community but rather about absorbing public funds to continue research.

*"If we consider that our research is important, and we ask for money for that because we need money, less than positions, but we still need money, and we ask societies, companies, or governments to fund our research, we should demonstrate that this research is important. I know that it is important but not enough that I know, I have to demonstrate it. The only way to demonstrate that our research is important is to show an interest among the society." (Interviewee J)*

From the resource-based view, the survival and performance of an entity strongly depends on its ability to leverage distinctive capabilities that lead to competitive advantages [39]. In the context of science, these capabilities are translated to research output for each individual and, thus, their survival in their scientific community. In our exploratory study, disseminating research through formal and informal mechanisms first and foremost serves to create an entry barrier for other individuals, which increases the likelihood of career survival of established scientists.

Taking the attention from macro-level factors such as institutions and economy, individuals develop their own survival mechanisms to sustain themselves in the science industry. Capturing value from science strongly influences individuals' survival. However, the big question arises here: should academic survival be the main motivator of utilizing value capture mechanisms?

### 4.3.2. Scientists' Ego-Identity Status Validation

The second category of underlying reasons for the perceived importance of subjective exchange value is scientists' validation of their own ego-identity status. We found that two types of ego-identity validation processes have a direct impact on the utilization of value capture mechanisms among scientists: personal ego-identity status and professional ego-identity status. Personal ego-identity refers to the individuals´ own definition of "who I am," while professional ego-identity status explains both one's awareness of being an employee doing a particular job and one's identification with its own group and social categories to which s/he relates by means of her/his job [40].

Recognition by the scientific community and then society is found to be one of the main pillars in ego-identity status validation when it comes to using value capture mechanisms. Especially in employing informal mechanisms, scientists' work being recognized by the public was the main antecedent.

> *"They [book presentations] should help people to get to know my name, and they should get people to know my research, "Oh, I did a book presentation," to people." (Interviewee I)*

> *"It is important that the public knows what research is doing or what it can actually do. Yeah, it is always tricky to find something that we could present there because many topics are not good to, yeah, as I said, grab the attention within a few seconds or minutes. But it is always very nice to go there and talk to mainly kids or teenagers." (Interviewee E)*

Our data shows that appearing in public places, giving public talks, presenting research to the public, and appearing in the media satisfied individuals' desire for social approval. The reason behind these types of behavior are explained as normative social behavior of beings: "To the extent that injunctive norms are based on individuals' perceptions about social approval, an underlying assumption in the influence of injunctive norms is that behaviors are guided, in part, by a desire to do the appropriate thing" [41]. For example, an interviewee claimed the reason why publishing a book was considered to be valuable is the "ego-boosting" effect of the action.

> *"[ . . . ] And then I thought, okay maybe for my habilitation[3] I can really start writing a book, and I've already a few, very big book chapters. [ . . . ] this is like an ego booster for me, to know that... yeah, I can honestly say this, [ . . . ]... because it's an intellectual challenge." (Interviewee C)*

Another important factor that is strongly associated with an ego-identity status is self-efficacy. Self-efficacy is defined as a personal judgement of "how well one can execute courses of action required to deal with prospective situations" [42]. Moreover, according to Cervone [43], individuals actively evaluate the relation between their perceived skills and the demands of tasks when thinking about their capabilities for performance. We observe various elements of self-efficacy when scientists explain their motives for involvement in dissemination activities.

> *"But by putting words on the piece of paper, you have to make sure that you are certain about what you are writing. So I do a lot of research to make sure that what I am writing is waterproof... is watertight, is foolproof [ . . . ]And by doing this, I also educate myself a lot, because I read about things that I, otherwise, maybe would not read, just to be really sure, and this gives me a lot of satisfaction because I am constantly educating myself and I really enjoy writing my own paper, or correcting somebody else's paper, to make sure that the structure is really perfect, because I really enjoy" (Interviewee C, formal mechanism)*

> *"Probably, because it is just again a skill training for presentations but, finally, if you get to the audience, the testimony then you have to show them that you are the man for the project. That could be quite good." (Interviewee G, informal mechanism)*

Hence, some scientists apply various types of dissemination mechanisms to perform a self-evaluation of their performance and their capabilities. In previous studies, self-efficacy was found to motivate scientists to perform research. Bandura [44] claims that research done by faculty needs noticeable creativity and scientists' motivation built on a strong sense of efficacy that their efforts are considered to be successful, which also depends on the field-specific demands. In our case, self-efficacy was found to be highly relevant in disseminating the research results.

### 4.3.3. Societal Impact

In the context of science, we tend to assume having a societal impact is potentially an antecedent for implementing value capture mechanisms. However, in our data, we found limited evidence of

---

[3] A habilitation is the highest qualification level issued by universities and is a requirement for full professorship in many European countries.

this. The desire to make a societal impact was not a primary driver for our study participants when answering the question of "why do you use certain mechanisms?".

Societal impact is mainly being driven by individuals' personal belief in research as a public good. In this vein, the outcome of research must directly benefit society. In our dataset, two scientists reported using media to create societal impact.

> "[ … ] So, to bring awareness to the public. But the content, what we are actually doing. Well at the end, it is public money that we use. And yeah. Especially when there are elections like there have been now. There might be changes in how big the share that goes to research. And then if people have no clue what research is actually doing for them, they cannot understand why they should give us a share." (Interviewee E)

## 5. Conclusions and Implications

This section briefly summarizes our six major results and discusses emerging theoretical contributions as well as practical implications for scientists, university managers, policy makers, and research funders.

First, scientists use dissemination mechanisms to capture value from the scientific knowledge they have produced. By disseminating their scientific knowledge, scientists empower academic and non-academic actors (e.g., the general public, firms, and policy makers) to capture use value from their knowledge if users pick it up. Second, the realized exchange value consists not only of a monetary dimension as conceptualized in previous studies, [2] but also of a subjective dimension that includes social recognition, reputation, and the validation of the ego-identity status. This finding is in line with prior research indicating that softer factors such as access to knowledge, reputation, or other non-monetary rewards might represent resources and, thus, value, on its own [15]. Third, the realized subjective exchange value is considered as valuable due to scientists' needs. Figure 6 illustrates this relationship. While the objective exchange value (direct or indirect monetary rewards) serves primarily to satisfy scientists' needs for academic survival, it also satisfies ego-identity needs. Receiving a meaningful grant can, for example, provide scientists with the desired funding to increase the chance of academic survival, while, at the same time, it pushes scientists' self-efficacy needs. Furthermore, while the realized exchange value can be clearly differentiated into objective and subjective rewards, the rewards' effect on satisfying needs is subjective (indicated by the blurred line in the needs' pyramid in Figure 6). Based on our data, we argue that some scientists ascribe a different utility to different types of rewards (e.g., while some scientists might ascribe a high utility to social recognition or salary increase, others might not). This perception might also be driven by disciplinary differences in what is considered in the evaluation scheme for scientists.

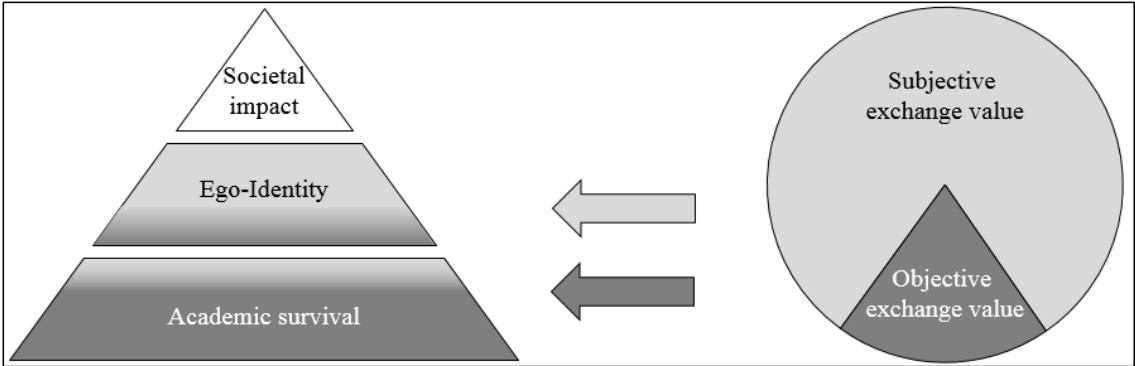

**Figure 6.** How the subjective and objective value satisfy scientists' needs. Source: own illustration. Please note that the fraction of the subjective versus objective value does not indicate a specific ratio.

Fourth, based on the previously mentioned findings, we can say that, not only traditional mechanisms (e.g., patenting, licensing) can be seen as value capture mechanisms, but rather all



kinds of dissemination mechanisms are able to realize exchange value—both objective and subjective exchange value—to different degrees. Thus, the overall picture of value capture mechanisms and realized exchange value needs to be considered as being more complex than previously assumed. Fifth, based on our results, neither the subjective nor the objective exchange value triggered societal impact. Surprisingly, societal impact was not explicitly mentioned as an underlying need. This becomes especially critical regarding the universities' Third Mission efforts. This might be because societal impact moves more into the background when compared to academic survival and ego-identity needs, so that the effects of the realized exchange value are not consciously observed. Sixth, the findings of this exploratory study emphasize the importance of considering individual-level factors when researching value creation and value capture processes, which is in line with Lepak, Smith, and Taylor [3]. Individuals, in this case, scientists, are the ones deciding to further engage in value creation processes or which value capture mechanism to apply. Hence, their cognitive and emotional processes play an important role in the overall scientific system. Our findings underline the importance of not only paying attention to macro-level factors, but also, considering the micro-foundations of scientific knowledge production, as individuals (e.g., scientists) are important decision makers.

These findings lead to three theoretical contributions. First, our findings contribute to the understanding of value capture in science while focusing on the dissemination mechanisms, the realized exchange value, and the circular relationship with engagement to create value. If value capture rationales are applied in the context of scientific research, realized exchange value cannot only be considered in monetary terms. This would lead to an insufficient driver to further engage in value creation (i.e., scientific knowledge production). For scientists, the largest fraction of realized value is subjective. Since the realized exchange value influences the anticipated exchange value [3], and this is a major driver to engage in knowledge production, it is critical to assess whether the realized exchange value is able to satisfy the underlying needs. Our findings uncover these needs (i.e., academic survival, ego-identity, and societal impact) that influence the individual perceived utility from the realized subjective exchange value. In turn, this individual utility, is influenced by the personal needs' structure and environmental factors such as disciplinary habits. Support for these findings can be found in the work motivation literature pointing out the importance for scientists to meet basic needs before their strong need for self-actualization can be pursued [12,32]. In addition to this literature stream, taking the value capture rationale as point of departure allows us to propose a categorization of value capturing mechanisms in the context of science. This also adds to the literature on knowledge production processes by shedding light on underlying reasons for producing and disseminating scientific knowledge and, thus, overcome the struggle of transferring scientific knowledge into practice [21]. This understanding helps design more beneficial negotiation opportunities, which leads to successful exchange and underlies the central role of universities in the knowledge production process [45]. This, in turn, is essential for the realization of exchange value (captured by the knowledge producing scientist), as well as the use value (captured by knowledge-using actors such as the general public). Please see Figure 7, which summarizes our contributions toward the understanding of value capture in science.

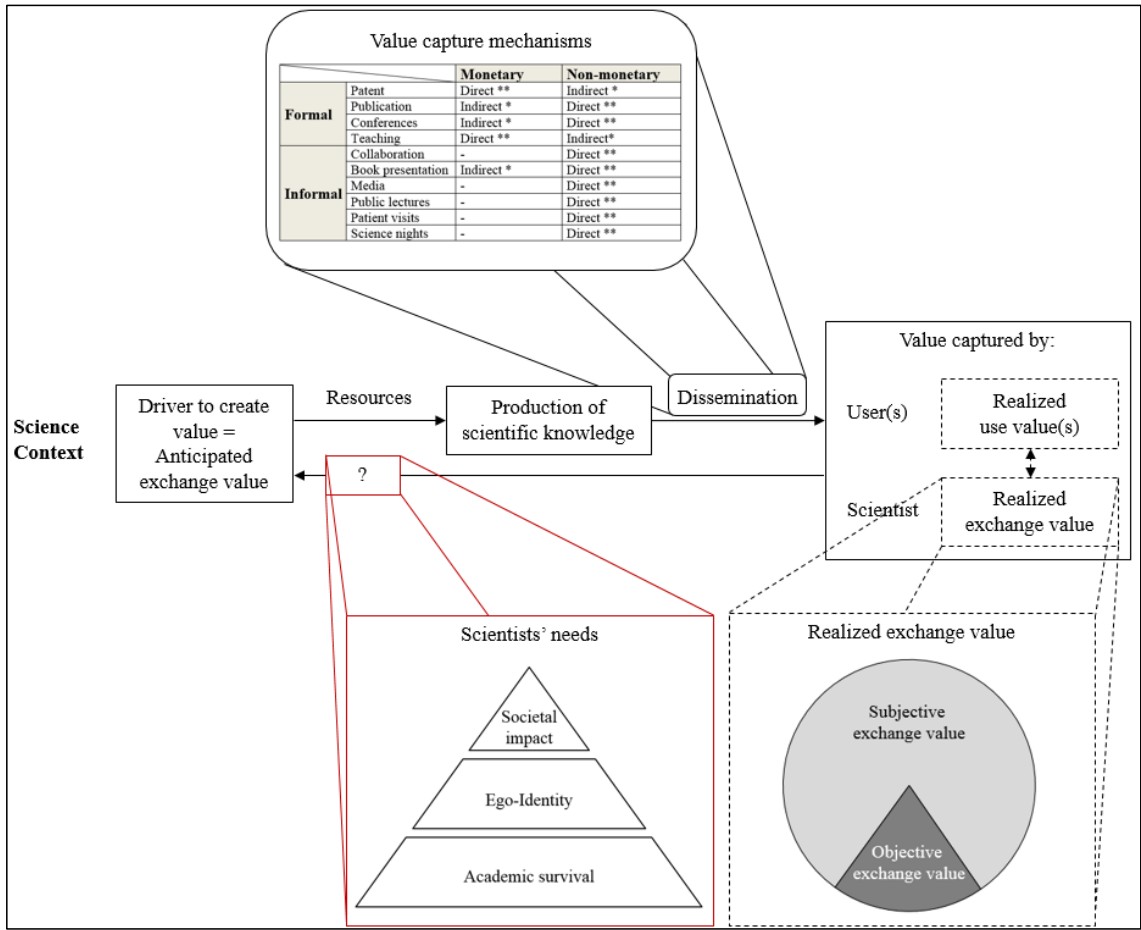

**Figure 7.** Conceptual model of value capture in science. Source: own illustration.

The second theoretical contribution adds to the discussion about the scientist's social engagement and contribution to Open Science, Third Mission, and the Triple and Quadruple Helix. Open Science, that is open knowledge production (e.g., citizen science) and open knowledge dissemination (e.g., open access journals) [46], can be described by its degree of openness, based on the possibility to participate and on the disclosure of intermediate inputs [46]. There is an increasing amount of evidence that the dissemination of scientific knowledge in terms of Open Science, i.e., sharing, reusing, recombining, and accumulating knowledge is more rewarding for individuals, institutes, a research field, and organizations compared to disclosed knowledge [47,48]. Discussing this reward against the background of differentiating use value and exchange value allows for a more precise incentivization for value creation and a better understanding of value distribution. Considering scientific knowledge as a commodity good allows us to disconnect realized use value from the exchange value. If the knowledge is freely available, each additional user of the knowledge increases the accumulated use value without increasing the exchange value—if the exchange value would be purely objective. But considering the subjective dimension of the exchange value opens opportunities to save scientists a part of the value—without decreasing the value captured by users. For example, the citation system recognizes this relationship. While citations have no costs for the user, the subjective exchange value for the scientists positively influences their external performance assessment. By making the subjective value visible, it becomes a currency that scientists can use to meet their need for academic survival. This, in turn, leads to further engagement to create value that can be captured as use value for societies, organizations, and individual users. In line with the understanding of generative appropriability [20], users of scientific knowledge can then create value on their own (e.g., by contributing solutions to deal with today's grand challenges). In terms of open knowledge production, the depicted findings may be

able to lower barriers to openly collaborate and share data or intermediate results, but also to involve the public in science processes to increase their scientific literacy. If scientists can be sure of capturing an appropriate piece of the value cake, their fear of not meeting their needs for academic survival and ego-identity can be decreased. Understanding subjective exchange value as a commodity allows for this inclusion of more actors in the value creation process, since they do not need to split the realized subjective exchange value. This is because the subjective exchange value depends on their own utility function. Simultaneously, sharing scientific knowledge among other scientists makes problem solving more likely and efficient [49]. Hence, it reduces the necessary effort and resources (e.g., time) to create use value (e.g., knowledge). Concluding, based on our findings, we argue that openly creating and disseminating knowledge does increase the use value of scientific knowledge (which is a major aim of Third Mission activities), but also the realized exchange value. Nevertheless, appropriate ways of making this subjective value more tangible are required.

Third, our findings contribute to the recent discussion on the influence of monetary and non-monetary rewards for knowledge workers by highlighting the importance of non-monetary reward systems driven by intrinsic motivations [50]. Our results indicate that salary supplements in any form can be considered as monetary rewards for engaging in Third Mission activities. However, they distinguish between the influence of basic salary versus salary supplements, which is how Frey and Neckermann [51] differentiated money and awards in their work. According to these authors, while money may bring recognition and status, awards are more effective. This is due to the fact that monetary rewards are not publicized, and knowledge on differences in basic salary is restricted to few, if any, close colleagues. In line with this, and the scientists' need a pyramid (see Figure 5), this provides a potential explanation why the basic salary might not function as monetary reward driving scientists to engage in Third Mission activities, while the effect of awards (e.g., research funding) is undeniable. Likewise, dissemination activities that support universities' Third Mission such as public engagement do (for now) limit the captured exchange value as long as the scientists' needs pyramid remains disregarded.

Our findings also bear important practical implications. For scientists, the awareness of this subjective exchange value might help in recognizing the value they receive from their work. The awareness that subjective exchange value is dependent on the individual utility function implies more control and options to receive a value in exchange for the knowledge production process. Furthermore, it allows a more precise estimation of the anticipated exchange value. In other words, being consciously aware of the subjective exchange value increases the actual exchange value scientists receive in exchange for their knowledge. This might increase their willingness to engage in value creation processes in the first place and affects their perceptions of competition arising from Open Science policies. Since the size of the use value is not directly related to the scientist's exchange value captured, new negotiation potential can be exploited.

For policy makers, research funders and university managers of these findings highlight a responsibility and a chance to change the current practices of scientific knowledge production and dissemination. First, the indication that scientists' willingness to create value is mainly driven by their need for academic survival is alarming. Although we do not want to draw any conclusions on scientists' performance, the current incentive system evokes pictures of gladiatorial combat. While many scientists drop out of the system due to a lack of objective exchange value (i.e., money, job), the subjective exchange value is also hardly considered valuable by outsiders. Therefore, new metrics that account for the subjective value and make it visible to outsiders are needed. Whether objective or subjective exchange value is realized seems to strongly depend on the dissemination strategy. However, the type of exchange value is currently not related to the quality of the scientist's work or the created use value. Policy makers should, therefore, be aware of the relationship between value creation, value capture, and the scientists' underlying needs. Second, scientists who contribute to creating substantial use value might currently drop out of academia due to a lack of sufficient objective exchange value, which is needed to survive in their academic career. Consequently, universities with a strong focus on

Third Mission and Triple/Quadruple helix efforts should particularly pay attention to avoiding such drop-outs. From a value capture perspective, scientists may need to be considered as entrepreneurs engaging in scientific knowledge production rather than employees. Third, the argument that salary is not directly linked to knowledge production and, hence, value capture, opens negotiation potential for appropriate payment that reduces the need to struggle for academic survival. Policy makers can increase the anticipated subjective exchange value to trigger dissemination strategies that yield higher value for society, which accomplishes the underlying aim of the Third Mission. Lastly, we highlight the need for developing and implementing novel capability building activities to raise scientists' awareness about different value capture strategies, their consequences, and relevant boundary conditions. We believe this is particularly important in training junior scientists and recommend integrating related discussions into the design of PhD programs. Building on recent insights on innovators' preferences for long-term engagement with scientists to collaboratively develop solutions for future, yet unknown problems [52], such capability building activities may need to particularly pay attention to value capture strategies in the context of science-based innovation. Ultimately, this can increase the share of knowledge being picked-up for innovations and, thus, create a sustainable societal impact.

## 6. Limitations and Future Research

This study has limitations that will hopefully motivate future research efforts. First, we apply an exploratory qualitative approach. As a next step, a large-scale validation study is required to test strategic patterns and contingencies influencing the scientists' strategic selections for appropriate value capture mechanisms. Identifying direct and indirect effects of different dissemination mechanisms on use and exchange value might provide deeper insights for scientists and policy makers. The simultaneous assessment of the consequences of certain mechanisms for the use value and the exchange value can provide meaningful insights for the creation of future incentive structures to foster Open Science and Third Mission. Second, while this study provides a first overview on a formal and an informal dissemination mechanism, future research would provide additional insights by further differentiating these mechanisms. For example, publications might vary regarding their degree of accessibility to the public (closed vs. open access), which leads to different degrees of use value or they might vary in terms of recognition by the research community (e.g., a publication in a top-tier journal might yield a higher exchange value). Third, this study's sample covers a large variety of different nationalities and scientists at different career stages. However, with the exception of some participants from humanities and social sciences, most participants come from fields related to biomedical scientific disciplines. This disciplinary concentration was suitable to observe heterogeneity in the applied value capture strategies. However, the observed differences between scientists from this field compared to scientists from the humanities and social sciences require further studies focusing on the contingencies resulting from research field related differences. For example, evaluation schemes for tenure positions might vary and, consequently, affect the value scientists capture from (un-)recognized dissemination activities. Diversity in terms of the nationalities of the participants is considered as less limiting (compared to their disciplines) due to the high levels of mobility among scientists and an increasing homogeneity regarding dissemination strategies across the world (e.g., publications in the same publishing houses). Fourth, our sample of study participants mainly consisted of scientists without a permanent position. Considering the importance of academic survival expressed by the scientists in our sample, we urge future studies considering scientists that already have a permanent position. It would be highly interesting to investigate how the scientist's needs and, consequently, what they consider as valuable, change with this event. It is very likely, that there is also an effect on the selection patterns for value capture mechanisms and different valorizations for objective and subjective exchange values. Fifth, the explorative nature of the research, and semi-structured interviews limit researchers in discovering factors if they do not appear during the course of interview. Future research needs to address other relevant behavioral, institutional, and field-specific factors that might influence knowledge dissemination mechanisms by scientists. Sixth, in line with this, we call for future research

to investigate potentially conflicting expectations toward the scientists' dissemination activities on the organizational and discipline-level. Especially when encouraging scientists' engagement in Third Mission activities, this might indirectly lead to individual-level resource allocation conflicts. Lastly, we want to point out that our paper makes no claims regarding scientists' research performance.

**Author Contributions:** Conceptualization, S.B., K.B., and M.P. Methodology, S.B., K.B., M.M., and M.P. Validation, S.B., K.B., M.M., and M.P. Formal analysis, M.M. and S.B. Investigation, S.B. Resources, M.P. Data curation, M.M. and S.B. Writing—original draft preparation, S.B., K.B., M.M., and M.P. Writing—review and editing, S.B., K.B., M.M., and M.P. Visualization, S.B. Supervision, M.P. and K.B. Project administration, S.B. Funding acquisition, M.P.

**Funding:** The Austrian National Foundation for Research, Technology and Development, grant for Open Innovation in Science Center I funded this research.

**Acknowledgments:** We acknowledge the valuable insights from the anonymous reviewers at the Academy of Management Annual Meeting 2018, the DRUID Society 2018 and the reviewers' comments during the review process.

**Conflicts of Interest:** The authors declare no conflict of interest.

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
