# Peer review of "The Value of Scientific Knowledge Dissemination for Scientists—A Value Capture Perspective"

_publications, doi:10.3390/publications7030054_

Reviewer 1 Report

This is a well-written manuscript, I applaud the authors' efforts to contribute to the field of knowledge production emphasizing the notion of scientists' value capture principles (Value of Scientific Knowledge) 

The theoretical background of the paper is adequate, the methodological design is suitable for an exploratory study, the analytical approaches appear to be appropriate, the presentation of findings is coherent, the limitations of the study are identified and the potential for future research is discussed.

In order to further improve the quality of this paper (specifically the background), I would suggest that recent studies on knowledge production processes (conceptual schemes of knowledge production) may be included. There have been numerous studies, that are directly associated with the epistemological foundations of knowledge production, as well as the structural divides of knowledge generation processes that form communities of scholars constituted by different intellectual camps. In my view, your article adds value to such former research endeavors on the conceptual scheme of knowledge production by exploring the scientific value principles, and the intellectual utility of knowledge practices.

Author Response

Comments and Suggestions for Authors

This is a well-written manuscript, I applaud the authors' efforts to contribute to the field of knowledge production emphasizing the notion of scientists' value capture principles (Value of Scientific Knowledge).

The theoretical background of the paper is adequate, the methodological design is suitable for an exploratory study, the analytical approaches appear to be appropriate, the presentation of findings is coherent, the limitations of the study are identified and the potential for future research is discussed.

In order to further improve the quality of this paper (specifically the background), I would suggest that recent studies on knowledge production processes (conceptual schemes of knowledge production) may be included. There have been numerous studies, that are directly associated with the epistemological foundations of knowledge production, as well as the structural divides of knowledge generation processes that form communities of scholars constituted by different intellectual camps. In my view, your article adds value to such former research endeavors on the conceptual scheme of knowledge production by exploring the scientific value principles, and the intellectual utility of knowledge practices.

Dear Reviewer 1,

thank you very much for the positive and constructive feedback on our manuscript and the chance to revise our work. In particular, we appreciate directing our attention to the literature on knowledge production processes. While the main objective is to understand the scientists individual value captured from disseminating her/his scientific knowledge, we also identify and suggest reasons for engaging in the knowledge production in the first place as you rightly pointed out. Your suggestion was key to better embed this additional part of the findings into existing literature (beyond value capture and motivation literature). Therefore, we included arguments and references to this literature in section 1 and 5 outlining how our individual-level findings on why scientists engage in the knowledge production in the first place nurture the knowledge production and transfer process. Thanks again very much for this valuable comment. Please find a list of the additional references below. Again, thank you very much for your support in moving this piece of research forward.

Selected new references:

Godin, Benoit, and Yves Gingras. "The Place of Universities in the System of Knowledge Production." Research Policy 29, no. 2 (2000): 273-78.

Gornitzka, Ă…se. Science, Clients, and the State: A Study of Scientific Knowledge Production and Use. Enschede: University of Twente, 2003.

Van de Ven, Andrew H, and Paul E Johnson. "Knowledge for Theory and Practice." Academy of Management Review 31, no. 4 (2006): 802-21.

Reviewer 2 Report

The article “The Value of Scientific Knowledge – A Value Capture Perspective” addresses a topic that fits the scope of “Publications” journal, and has generated an extensive academic debate over the past years. The article analyses the scientific knowledge production and its dissemination to the knowledge-users. While the majority of the work produced on this topic focus the knowledge-users perspective to underline the relevance of the knowledge transfer mechanisms, this article proposes to do it using the perspective of the knowledge-producers. This focus on scientists, when analysing the knowledge production and dissemination, is fundamental do understand the whole phenomenon. As such, in my opinion it is an article that can be published in “Publications” journal.

Some recommendations to improve the article:

-      Generically, the article is well written, presents a good English level and has an adequate length. Nevertheless, it would benefit from a final reading to eliminate some minor errors in the text.

-      Figures and tables: source identification is missing.

-      Figure 3: according to the text explanation the title of phase 2 and phase 3 seem to be swapped. Phase 2 should be represented after phase 1, and not in the end.

-      The analysis results present an interesting focus on the differentiation of health sciences and humanities and social sciences, but only in the results section. This should also be reflected in the theoretical background section. The several knowledge dissemination mechanisms existent, and identified by the authors, do not have the same application in health sciences and social sciences: for instance, the use frequency of the different type of knowledge dissemination mechanisms is distinct, the type of knowledge-users may vary a lot (companies, third sector organizations, individual professionals making use of different types of knowledge – formal, informal and symbolic…), and adequacy issues. That surely  influences also the results readings.

-      It is interesting that the analysis presented differ health sciences from humanities and social sciences. But from the 11 interviews only 3 are from social scientists, which make it very hard to build strong conclusions. This situation should be addressed in section 3.3..

-      The article does not have a concluding section, as it is recommended by the “Publications” instructions for authors! Some changes should be done in the sections titles in order to reflect their real content: section 4, called “Analysis” by the authors, should be called “Results; and section 5 should be called “Conclusion and policy implications”.

Author Response

Comments and Suggestions for Authors

The article “The Value of Scientific Knowledge – A Value Capture Perspective” addresses a topic that fits the scope of “Publications” journal, and has generated an extensive academic debate over the past years. The article analyses the scientific knowledge production and its dissemination to the knowledge-users. While the majority of the work produced on this topic focus the knowledge-users perspective to underline the relevance of the knowledge transfer mechanisms, this article proposes to do it using the perspective of the knowledge-producers. This focus on scientists, when analysing the knowledge production and dissemination, is fundamental do understand the whole phenomenon. As such, in my opinion it is an article that can be published in “Publications” journal.

Dear Reviewer 2,

thank you very much for your encouraging evaluation of our manuscript and your constructive feedback. We are delighted by the chance to re-submit a revised version of our manuscript. Please see below a point-to-point response to your comments that in our opinion were substantial to increase this manuscript’s potential to contribute to understanding the value of scientific knowledge dissemination.

Some recommendations to improve the article:

Comment 1:

Generically, the article is well written, presents a good English level and has an adequate length. Nevertheless, it would benefit from a final reading to eliminate some minor errors in the text.

Response to comment 1:

Thank you very much. We sent the manuscript to a professional proofreading service and by this hope to have eliminated remaining errors.

Comment 2:

Figures and tables: source identification is missing.

Response to comment 2:

Thanks a lot, we have added the according information to each figure. Figure 1 and 2 were adapted from Bowman and Ambrosini (2000) while all other figures were created by the authors.

Comment 3:

Figure 3: according to the text explanation the title of phase 2 and phase 3 seem to be swapped. Phase 2 should be represented after phase 1, and not in the end.

Response to comment 3:

Thank you very much for bringing this to our attention and your careful reading. We have corrected the mistake.

Comment 4:

The analysis results present an interesting focus on the differentiation of health sciences and humanities and social sciences, but only in the results section. This should also be reflected in the theoretical background section. The several knowledge dissemination mechanisms existent, and identified by the authors, do not have the same application in health sciences and social sciences: for instance, the use frequency of the different type of knowledge dissemination mechanisms is distinct, the type of knowledge-users may vary a lot (companies, third sector organizations, individual professionals making use of different types of knowledge – formal, informal and symbolic…), and adequacy issues. That surely  influences also the results re(adings.

Response to comment 4:

Thank you very much for this valuable suggestion. We fully agree that the selection of a dissemination mechanisms is influenced by the field. We have now included the field- and discipline-dependency throughout the manuscript, specifically in theoretical background (section 2.1.), section 5 (discussion) and section 6 (future research). Moreover, in relation to your next comment, we agree that future studies need to specifically address disciplinary differences. Nevertheless, our findings recognize the influence of field-specific evaluation schemes that help scientists to achieve position-related goals such as a tenured position. Hence, while we cannot outline specific field-related differences due to our sample, we propose that evaluation schemes differ between scientific fields and thus, different dissemination activities yield different exchange values for scientists in different fields. Thanks again for helping us recognize that this needs to be emphasized throughout the manuscript.
Comment 5:

It is interesting that the analysis presented differ health sciences from humanities and social sciences. But from the 11 interviews only 3 are from social scientists, which make it very hard to build strong conclusions. This situation should be addressed in section 3.3..

Response to comment 5:

Thank you very much. We agree and emphasized this in section 3.3.: “ Although our interviews were selected from different scientific fields with different nationality backgrounds, the issue of generalizability has always been present in doing our qualitative research. For example, while the scientists were from different disciplines, the sample comprises a smaller fraction of scientists from social sciences and humanities (each n=2)”. This is in line with the outlined limitation: “Third, this study’s sample covers a large variety of different nationalities and scientists at different career stages. But with the exception of some participants from humanities and social sciences, most participants come from fields related to biomedical scientific disciplines. This disciplinary concentration was suitable to observe heterogeneity in the applied value capture strategies. But the observed differences between scientists from this field compared to scientists from the humanities and social sciences require further studies focusing on the contingencies resulting from research field related differences.”

While we also agree that it would be nice to reveal more detailed field-related idiosyncrasies we rejected this idea due to the protection of the participants’ anonymity. Nevertheless, we would like to let you know that the scientists from the field of “health science” have very diverse backgrounds.  

Comment 6:

The article does not have a concluding section, as it is recommended by the “Publications”instructions for authors! Some changes should be done in the sections titles in order to reflect their real content: section 4, called “Analysis” by the authors, should be called “Results; and section 5 should be called “Conclusion and policy implications”.

Response to comment 6:

Thank you very much, we fully agree. We have changed section 4 accordingly to “Analysis and Results” and section 5 “Conclusion and Policy Implications”.
Again, thank you very much for your support in this review-process which we highly appreciate.

Reviewer 3 Report

The paper is raises a very important topic, both from organizational and individual perspectives. We recommend to pay more attention when discussing the two facets and to comment specifically for the two situations. As well as, viewing the two situations in relationship is also relevant. For instance, the paper mentions the Third mission (an organizational target/framework). How would this influence the personal variables of disseminating research?

Considering a wider macro-level framework, theories related to Quadruple Helix stimulating cooperation among the members involved might lead o new insights. Thus, not only the Third mission of an organizaTion is relevant but also the specific dynamics of a Triple/Quadruple/... Helix consortium.

Methodology is well grounded and documented. Nevertheless, limits have to be considered more specifically. For instance, the research practices are influenced by the home-departments of researchers and wide differences exist among universities and counties. Domain-related variations might be also observed.

We recommend to pay attention to the presentation of the main results in some sections of the papers. Qualitative studies generate more nuanced findings than the ones presented in the introductory section and they present several facets of each item considered. The paper tends to select certain aspects as paramount (for instance cognitive factors are stressed while subjective/human aspects are almost dismissed in the begining while towards the end it seems to suggest that the are paramount).

Mechanisms - the paper tends to put more stressed on formal mechanisms of disseminations.  We recommend to expand the informal aspects if data is available the analysis of informal mechanisms. They are probably more subjective in nature, therefore more difficult to grasp. It would be also relevant to mention he degree in which the interviewees are aware of the monetary and non-monetary benefits, or these aspects are identified by the authors rather than during the interviews.

Objective vs.subjective exchange - Please explain how the percentages associated to Figure  have been calculated. How could the ratio between the two approaches be determined?

Discussions could be extended considering other variables, such as external vs.internal pressures on scietists, or use of open-access versus closed-access dissemination channels.

Author Response

Comments and Suggestions for Authors

Dear Reviewer 3,

Thank you very much for your positive review of our manuscript. We very much appreciate your constructive feedback and hope that the revised version addresses your concerns. We believe that the changes due to yours and the other reviewers’ comments helped a lot to improve the clarity and positioning of our manuscript. Please see below a point-to-point response to your comments.

Comment 1:

The paper is raises a very important topic, both from organizational and individual perspectives. We recommend to pay more attention when discussing the two facets and to comment specifically for the two situations. As well as, viewing the two situations in relationship is also relevant. For instance, the paper mentions the Third mission (an organizational target/framework). How would this influence the personal variables of disseminating research?

Response to comment 1:

Thank you so much for this enlightening comment. It helped us to refine the positioning of our manuscript by distinguishing much clearer between the individual-level (what we are looking at) and the organizational-level (where our findings have implications, i.e., for institutions to designing incentive schemes for scientists to disseminate their research). Therefore, we have reworked the title, abstract and introduction to clearly outline what this study addresses: How do individual scientists capture value from their scientific knowledge production and dissemination activities? Hence, we emphasize the scientists’ individual-level. Again, thank you for recommending the clearer distinction between the individual- and organizational-level as well as the relation. We believe that this was key to bridge the research question and empirics (individual-level) with our major contributions (organizational / policy-level).

Comment 2:

Considering a wider macro-level framework, theories related to Quadruple Helix stimulating cooperation among the members involved might lead to new insights. Thus, not only the Third mission of an organizaTion is relevant but also the specific dynamics of a Triple/Quadruple/... Helix consortium.

Response to comment 2:

Thanks a lot for suggesting the inclusion of the Triple / Quadruple Helix. We agree that our findings might have implications for both concepts. We new relate our findings not only to the goals of the Third mission concept but outline that Triple and Quadruple Helix aim for similar goals. We therefore have included additional references that underline this relationship and understanding:

1. Carayannis, Elias G, and David FJ Campbell. "Triple Helix, Quadruple Helix and Quintuple Helix and How Do Knowledge, Innovation and the Environment Relate to Each Other?: A Proposed Framework for a Trans-Disciplinary Analysis of Sustainable Development and Social Ecology." International Journal of Social Ecology and Sustainable Development (IJSESD) 1, no. 1 (2010): 41-69.

2. Zawdie, Girma. "Knowledge Exchange and the Third Mission of Universities: Introduction: The Triple Helix and the Third Mission–Schumpeter Revisited." Industry and Higher Education 24, no. 3 (2010): 151-55.

Comment 3:

Methodology is well grounded and documented. Nevertheless, limits have to be considered more specifically. For instance, the research practices are influenced by the home-departments of researchers and wide differences exist among universities and counties. Domain-related variations might be also observed.

Response to comment 3:

Thank you very much for directing our attention to this issue. We fully agree. Your comment also triggered a discussion on conflicting expectations from scientists' departments and disciplines. Consequently, we have included, among others, two limitations that address your comment: “Fifth, the explorative nature of the research, and semi-structured interviews limits researchers in discovering factors if they do not appear during the course of interview. Future research needs to address other relevant behavioral, institutional, and field-specific factors that might influence knowledge dissemination mechanisms by scientists.Sixth, in line with this we call for future research to investigate potentially conflicting expectations towards the scientists’ dissemination activities on the organizational and discipline-level. Especially when encouraging scientists’ engagement in Third Mission activities this might indirectly lead to individual-level resource allocation conflicts.” Moreover, we included throughout the manuscript that the value perceived from scientific dissemination mechanisms is also influenced by field-specific expectations. Fore example, while some fields might recognize books more than peer-reviewed publications when it comes to scholarly evaluations, others might consider peer-reviewed articles more valuable.

We hope that we are now more precise and sharp in limiting the contributions that our manuscript is able to make to understanding the value of scientific knowledge dissemination.

Comment 4:

We recommend to pay attention to the presentation of the main results in some sections of the papers. Qualitative studies generate more nuanced findings than the ones presented in the introductory section and they present several facets of each item considered. The paper tends to select certain aspects as paramount (for instance cognitive factors are stressed while subjective/human aspects are almost dismissed in the begining while towards the end it seems to suggest that the are paramount).

Response to comment 4:

Thank you very much for raising this point. We now address this in a paragraph in the results section and in the limitation section as limitation of our qualitative data inquiry. Indeed we agree with you and tried to overcome this obstacle by applying an open coding procedures using semi-structured interviews that encourages the data to speak for itself. And in line with your observation, some facets might appear more often from interviews than others, we won't draw conclusions based on the quantity of facets but describing why, how and when a certain facet appears and what are the consequences. Thanks to your suggestion we were able to address this issue in the revised manuscript.

Comment 5:

Mechanisms - the paper tends to put more stressed on formal mechanisms of disseminations.  We recommend to expand the informal aspects if data is available the analysis of informal mechanisms. They are probably more subjective in nature, therefore more difficult to grasp. It would be also relevant to mention the degree in which the interviewees are aware of the monetary and non-monetary benefits, or these aspects are identified by the authors rather than during the interviews.

Response to comment 5:

Thank you for your constructive approach to this issue. We made our methodology section clearer in terms of how the data was collected and how much information was given to participants prior to the interviews. We included in the data collection part (section 3.1.) some clarifications to address this issue. The purpose of the interviews were to ask scientists to elaborate on different dissemination mechanisms openly and we dug into the reward mechanisms during the course of interview, however none of our participants mentioned the terms “monetary rewards” or “non-monetary rewards”. This categorization was a result of the second analytical steps in analysing the data. In addition, we tried to clarify that we do not claim that monetary rewards are unimportant but that these were less emphasized by the participants. We hope that the changes address your concerns appropriately, thanks again for sharing your concern. .

Comment 6:

Objective vs.subjective exchange - Please explain how the percentages associated to Figure  have been calculated. How could the ratio between the two approaches be determined?

Response to comment 6:

Thank you very much for raising this question. Please apologize for the confusion. The ratio is not calculated but should rather indicate symbolically that the largest part of the overall value captured from disseminating knowledge is subjective. We have indicated this in the caption of Figure 4 and Figure 6 to avoid readers’ confusion.  

Comment 7:

Discussions could be extended considering other variables, such as external vs.internal pressures on scietists, or use of open-access versus closed-access dissemination channels.

Response to comment 7:

Thank you very much and we fully agree with you - this would be extremely interesting. Unfortunately, our data does not allow to distinguish different options within the same dissemination mechanism (e.g., different types of scholarly publications). Hence, we included your very valuable and interesting thought in the future research section: “Second, while this study provides a first overview on formal and informal dissemination mechanism, future research would provide additional insights by further differentiating these mechanisms. For example, publications might vary regarding their degree of accessibility to the public (closed vs open access) leading to different degrees of use value or they might vary in terms of recognition by the research community (e.g., a publication in a top-tier journal might yield a higher exchange value).”

Considering the internal and external pressures we added another avenue for future research further below in the manuscript: “Sixth, in line with this we call for future research to investigate potentially conflicting expectations towards the scientists’ dissemination activities on the organizational and discipline-level. Especially when encouraging scientists’ engagement in Third Mission activities this might indirectly lead to individual-level resource allocation conflicts.” Again, thank you very much for pointing this out.

Again, thank you very much for your support in moving this piece of research forward.

Reviewer 4 Report

Dear Authors,

As a reviewer, I found your manuscript very interesting. it addresses an important subject and employed a good method to investigate it.

I see no serious flaw in your work, but I have some suggestions. firstly, as a reader, I needed to read your introduction several times to understand clearly what is your specific problem.

You started very soon by "third mission" in your abstract and then in your introduction, without providing a background for the reader. with no reference to this word in your title, or prior background, the reader confuses. similarly, your introduction is partly confusing. for this reason I suggest you to rewrite your introduction to make it easier for the readers to get the point.

I also recommend you to reconsider your title. I think it doesn't reflect the content correctly.

Also there is no managerial implication in your conclusion part. suggestions for future researches are short and insufficient and it lacks managerial implications. I strongly suggest you to make a section for your suggestions, both future researches and managerial.

I have no more comments. best of luck

Author Response

Comments and Suggestions for Authors

Dear Authors,

Comment 1:

As a reviewer, I found your manuscript very interesting. it addresses an important subject and employed a good method to investigate it.

Response to comment 1:

Dear Reviewer 4,

We very much appreciate the time and effort you have invested in formulating your very constructive comments. We feel encouraged about the chance to revise our manuscript and hope that we were able to incorporate your suggestions appropriately. We are confident that the changes made due to yours and the other reviewers’ comments helped to further increase the potential contribution of our manuscript. Please see below a point-to-point response to each of your comments.

Comment 2:

I see no serious flaw in your work, but I have some suggestions. firstly, as a reader, I needed to read your introduction several times to understand clearly what is your specific problem.

You started very soon by "third mission" in your abstract and then in your introduction, without providing a background for the reader. with no reference to this word in your title, or prior background, the reader confuses. similarly, your introduction is partly confusing. for this reason I suggest you to rewrite your introduction to make it easier for the readers to get the point.

Response to comment 2:

Thank you for pointing on the difficulties you had when reading the front-end of the manuscript. We reworked the title, abstract and introduction to make it clearer (and more consistent) what this paper investigates: what is the value that individual scientists receive (i.e., capture) from disseminating their scientific discoveries and how do they capture this value. We fully agree that we missed to specifically outline that our manuscript rather serves to inform how to realize “Third Mission” activities instead of being the starting point. Hence, we now emphasize the discussion on Third Mission in our implication section, and less in the front-end.

Our underlying rationale as we have emphasized it in the revised manuscript is that Third Mission is a concept addressing organizational level changes in terms of aiming to push universities towards increasing the societal impact of their actions. However, it is individual scientists that are embedded in the organization (i.e., university), that then need to choose mechanisms for disseminating their research. And considering that different dissemination mechanisms yield different levels of societal impact, we argue that based on our findings, universities, research funders, and policy makers can incentivize individual scientists to use dissemination mechanisms that create stronger societal impact (higher use value), as aimed for by third mission activities.

Comment 3:

I also recommend you to reconsider your title. I think it doesn't reflect the content correctly.

Response to comment 3:

Thank you very much. We have intensively discussed the title and hope that the new one reflects more clearly and consistently the content of the paper accurately: “The Value of Scientific Knowledge Dissemination for Scientists – A Value Capture Perspective”. By pointing both on the dissemination and the focus on the scientists’ individual-level we believe to better reflect our research question as well as the major contributions.  

Comment 4:

Also there is no managerial implication in your conclusion part. suggestions for future researches are short and insufficient and it lacks managerial implications. I strongly suggest you to make a section for your suggestions, both future researches and managerial.

I have no more comments. best of luck

Response to comment 4:

Thanks a lot for outlining an improved structure for the manuscript. In line with Reviewer 2 and the author guidelines of Publications we have changed the headings of the sections to better reflect their content. We now have a “conclusion and implications” section that outlines theoretical contributions as well as practical implications for researchers, policy makers, university managers and research funders. We hope that this rearrangement fits your suggestion. We have reworked the “conclusions and implications” section, mapping several insights that we believe contribute to achieving societal impact by understanding the value individual scientists capture from disseminating their scientific knowledge. Thereby, we have also added several managerial implications for university managers, policy makers and research funders. For example, there is a need for capability-building already at a very early career stage to support informed selection of different dissemination mechanism. Likewise, we have improved the structure of section 6 (Limitations and Future Research) and have added three additional ideas for future research. Again, thank you very much for your support in advancing this piece of research.

Round  2

Reviewer 2 Report

Overall, the authors answered all my previous recommendations. Thus, I have no further comments on this article.